# WHotLAMP: A simple, inexpensive, and sensitive molecular test for the detection of SARS-CoV-2 in saliva

David Ng[1], Ana Pinharanda[2]°, Merly C. Vogt[2,3]°, Ashok Litwin-Kumar[1], Kyle Stearns[4], Urvashi Thopte[1], Enrico Cannavo[1], Armen Enikolopov[7], Felix Fiederling[1], Stylianos Kosmidis[1], Barbara Noro[1], Ines Rodrigues-Vaz[1], Hani Shayya[1], Peter Andolfatto[2], Darcy S. Peterka[1], Tanya Tabachnik[1], Jeanine D'Armiento[4,5], Monica Goldklang[4,5], Andres Bendesky[1,6]*

1 Zuckerman Mind Brain Behavior Institute, Columbia University, New York, NY, United States of America, 2 Department of Biological Sciences, Columbia University, New York, NY, United States of America, 3 Howard Hughes Medical Institute, Columbia University, New York, NY, United States of America, 4 Department of Anesthesiology, Columbia University Irving Medical Center, New York, NY, United States of America, 5 Center for LAM and Rare Lung Diseases, Columbia University Irving Medical Center, New York, NY, United States of America, 6 Department of Ecology, Evolution and Environmental Biology, Columbia University, New York, NY, United States of America, 7 Independent Researched, New York, NY, United States of America

☯ These authors contributed equally to this work.
* a.bendesky@columbia.edu

☑ OPEN ACCESS

**Data Availability Statement:** All relevant data are within the manuscript and its Supporting Information files.

## Abstract

Despite the development of effective vaccines against SARS-CoV-2, epidemiological control of the virus is still challenging due to slow vaccine rollouts, incomplete vaccine protection to current and emerging variants, and unwillingness to get vaccinated. Therefore, frequent testing of individuals to identify early SARS-CoV-2 infections, contact-tracing and isolation strategies remain crucial to mitigate viral spread. Here, we describe WHotLAMP, a rapid molecular test to detect SARS-CoV-2 in saliva. WHotLAMP is simple to use, highly sensitive (~4 viral particles per microliter of saliva) and specific, as well as inexpensive, making it ideal for frequent screening. Moreover, WHotLAMP does not require toxic chemicals or specialized equipment and thus can be performed in point-of-care settings, and may also be adapted for resource-limited environments or home use. While applied here to SARS-CoV-2, WHotLAMP can be modified to detect other pathogens, making it adaptable for other diagnostic assays, including for use in future outbreaks.

## Introduction

The severe acute respiratory syndrome coronavirus 2 (SARS-CoV-2) is a novel coronavirus with high transmissibility that causes the Coronavirus Disease of 2019 (COVID-19) [1, 2]. Unlike SARS-CoV-1, where infectiousness is mostly restricted to the symptomatic phase [3], ~50% of SARS-CoV-2 transmissions occur 1–2 days before symptom onset or through people who never develop symptoms [4–6]. Thus, screening for symptoms is not enough to stop SARS-CoV-2 transmission [7].

**Funding:** This study was funded by Friends of the Zuckerman Institute and Columbia University and NIH grant GM112758 to PA. AB is a Searle Scholar (https://www.searlescholars.net/), Klingenstein-Simons Neuroscience Fellow (https://klingenstein.org/esther-a-joseph-klingenstein-fund/neuroscience/) and a Sloan Neuroscience Fellow (https://sloan.org/fellowships/). The funders had no role in study design, data collection and analysis, decision to publish, or preparation of the manuscript.

**Competing interests:** Patent application 63/088,694 related to this technology was filed by Columbia University, to facilitate that this technology be made widely available. AB became a member of the advisory board of Rapid Diagnostic Systems Limited for which he received options, after the conclusion of experiments and data analyses described here. This does not alter our adherence to PLOS ONE policies on sharing data and materials.

Testing, combined with contact tracing and social isolation, along with physical barriers such as face masks and distancing, became staple strategies to reduce community spread [8]. However, in many countries, viral spread has been difficult to contain. This is partly due to insufficient testing infrastructure, which leads to long delays in both access to testing and in obtaining test results. This lag greatly reduces the effectiveness of contact-tracing and isolation strategies [9, 10].

Despite the development of safe and effective vaccines against SARS-CoV-2, the threat of the virus remains high because of the logistical difficulties of global vaccination, limited supply of vaccine doses, and reluctance to get vaccinated [11]. Moreover, the emergence of SARS-CoV-2 variants that lower the protection conferred by natural or vaccine-induced immunity, suggests that testing will remain an important tool to reduce viral transmission [12–14]. Furthermore, without global vaccination coverage there is potential for future viral outbreaks.

Frequent testing using 'rapid' tests has been proposed as an effective strategy to survey the population and identify infectious people [9, 10], permitting a faster and safer return to pre-pandemic social and economic activities. A frequent testing strategy is effective if a test is: 1) rapid; 2) inexpensive; 3) simple to use (ideally self-administered for convenience and to minimize health-care resources); 4) sensitive enough to identify most infectious people; and 5) highly specific, so that when prevalence is low, most positives tests are true.

Initial tests to detect SARS-CoV-2 infection used deep nasopharyngeal swabs followed by RT-qPCR and were conducted by specially trained personnel [15, 16]. To increase testing capacity and reduce time to get a test result, point-of-care (POC) and home-based diagnosis using 'rapid' tests were developed to detect viral antigens from shallow nasal swabs [17]. While these tests provide quick results, even the most sensitive of antigen tests can only detect ~20,000 viral E gene RNA copies per microliter (μL) [18] and may miss up to 30% of people with the high viral loads associated with infectivity [18–28].

Other molecular 'rapid' tests detect viral RNA using an isothermal enzymatic reaction to exponentially amplify fragments of the genome [29–34], and some have been approved for emergency use for POC and at home [35, 36]. While this type of rapid test is sensitive, they are expensive and/or require specialized equipment (i.e. are not simple to use). Saliva offers several advantages over nasal and nasopharyngeal swabs for the early detection of SARS-CoV-2 infections: it 1) has a higher viral load than swabs early in an infection [37, 38]; 2) is easier than swabs to obtain from children, who are often anxious about the swabbing procedure; 3) requires fewer materials to collect, diminishing waste and reliance on resources that can be scarce. Several protocols to detect SARS-CoV-2 from saliva have been described [28, 39]. However, these approaches require toxic chemicals or specialized equipment (e.g. centrifuges, pipettes, thermocyclers) which make them impractical for POC, home testing, and other resource-limited environments.

Given these considerations, a fast, economical, easy to use test that is both sensitive, specific, and safe, is still required. To this end, we devised WHotLAMP, a rapid molecular test to detect SARS-CoV-2 viral RNA directly from saliva without the need for specialized equipment, with results obtained in 30 minutes. This test extracts RNA from saliva and uses an isothermal enzymatic reaction to amplify and colorimetrically detect SARS-CoV-2 RNA. WHotLAMP is inexpensive, highly sensitive and specific, making it ideal for frequent screening and detection of infectious individuals to limit the spread of SARS-CoV-2.

## Results and discussion

### A one-tube saliva test to detect SARS-CoV-2 RNA

To develop a simple procedure to extract SARS-CoV-2 RNA from saliva, we leveraged the nucleic acid binding properties of cellulose paper (e.g. Whatman filter paper [40]), and

molecular detection of SARS-CoV-2 RNA using loop-mediated isothermal amplification (LAMP), an enzymatic reaction that exponentially amplifies a target nucleic acid sequence at a constant temperature [41]. Inspired by work from Liu et al. [42], we sought to develop an inexpensive, sensitive and simplified test that did not require potentially dangerous chemicals, and would be suitable for frequent use at POC and adaptable for home use. We initially tested saliva spiked with naked SARS-CoV-2 control RNA and found that a short exposure of Whatman No. 1 filter paper to saliva, followed by brief washes, could capture sufficient SARS-CoV-2 control RNA to be detected in a LAMP reaction with primers directed against SARS-CoV-2 (Fig 1A). Amplification of the target sequence leads to a drop in pH, which is detected with a pH-sensitive dye as a color change from pink (negative) to yellow (positive) [32]. We next tested whether we could capture encapsulated SARS-CoV-2 RNA particles that were spiked into saliva, using Whatman paper and an established lysis buffer [40]. This procedure can

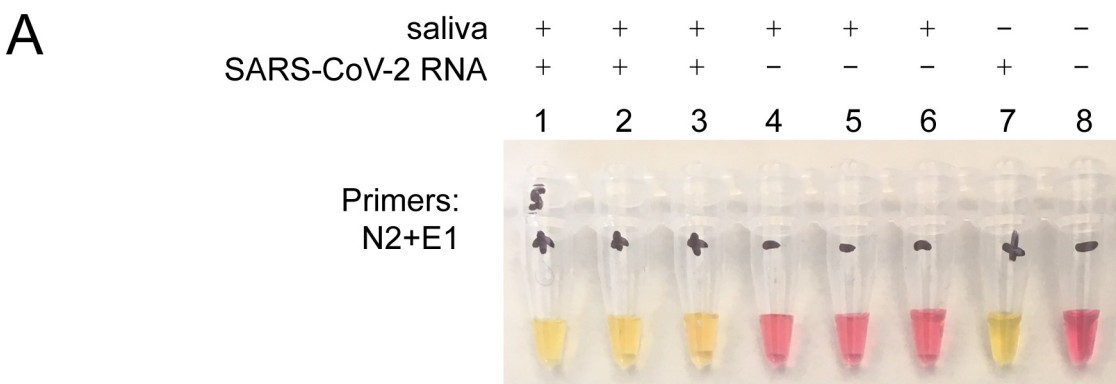

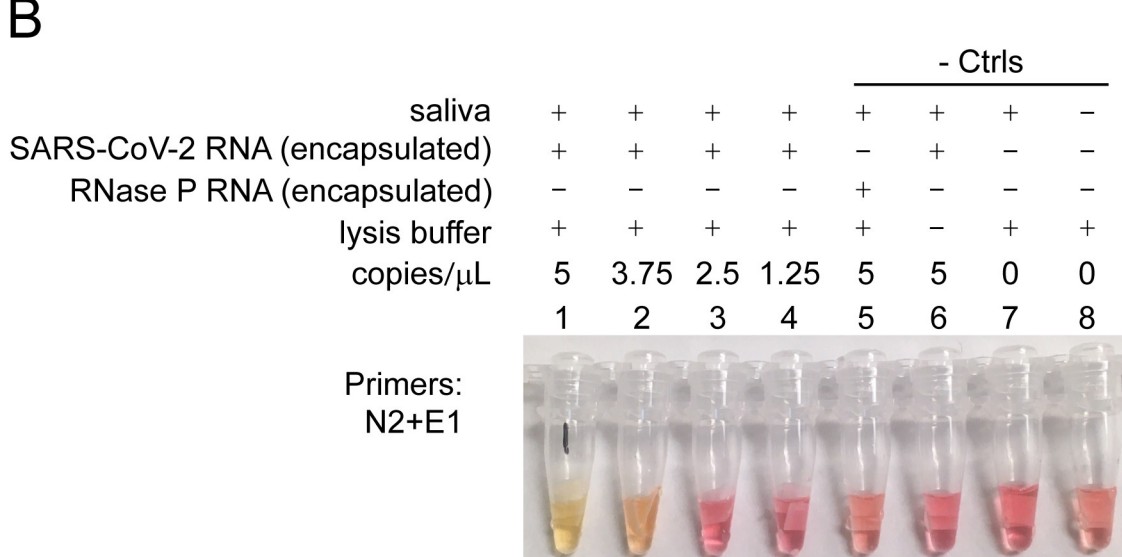

**Fig 1. Loop-mediated isothermal amplification (LAMP) detection of SARS-CoV-2 RNA captured from saliva using Whatman no. 1 filter. A**. Detection of naked SARS-CoV-2 RNA in saliva. Saliva with spike-in SARS-CoV-2 RNA (tubes 1–3), saliva without RNA spike-in (tubes 4–6), SARS-CoV-2 RNA added directly to LAMP reaction (tube 7), no template control (tube 8). **B**. Detection of encapsulated SARS-CoV-2 RNA particles in saliva (tubes 1–4); saliva with spike-in encapsulated RNase P RNA particles (tube 5); saliva with spike-in encapsulated SARS-CoV-2 RNA particles with no extraction treatment (tube 6), saliva alone with no spike-in (tube 7), and no saliva (tube 8). LAMP reactions used N2+E1 primers for detection of SARS-CoV-2 RNA. Concentrations are in copies per microliter of saliva.

**Table 1. Compatibility of materials with colorimetric LAMP.**

| Compound | Color change in LAMP (from pink) |
|---|---|
| Kwik-Sil™ silicone elastomer (World Precision Instruments) | N |
| Clear aquarium silicone (Aqueon) | Y |
| Neutral cure silicone (Dow Corning 737) | Y |
| Ultra-clear polyester plastic sheet (dipstick) | N |
| Glue stick (polyamide, Power Adhesives TEC Bond 7718) | Y |
| Glue stick (acrylic, Infinity Bond) | Y |
| Glue stick (high temperature, Allary) | Y |
| Gorilla glue (original, polyurethane) | Y |
| Crazy Glue (Loctite) | Y |
| Scotch permanent double-sided tape | Y |
| Rubber cement (Best-Test paper cement) | Y |

detect as few as ~4 SARS-CoV-2 particles per µL of saliva (Fig 1B). This level of sensitivity is notable, as it has been determined that 90% of COVID-19 patients carry more than 5 copies of SARS-CoV-2 per µL of saliva [28, 38]. These findings suggest that a strategy using Whatman paper is a viable approach for isolating RNA from SARS-CoV-2 virions in saliva.

Since the original extraction procedure used guanidine hydrochloride (a toxic protein denaturant), we sought an alternative that avoided toxic chemical components. We developed a saliva extraction procedure using only two components, a non-toxic RNA preservative (RNAlater™), and an endopeptidase, Proteinase K. To minimize handling of Whatman paper, we secured a piece of Whatman paper to the bottom of a 1.7 mL centrifuge tube using a small amount of Kwik-Sil™ silicone adhesive. Notably, Kwik-Sil™ did not interfere with the colorimetric pH indicator in the LAMP reaction mixture, whereas other adhesives we examined caused a color change in negative-control reactions without RNA (Table 1).

WHotLAMP can be performed entirely in a single 1.7 mL microfuge tube (Fig 2A). In this test, viral RNA is preserved using a non-hazardous RNA stabilizing solution and is extracted by a brief Proteinase K digestion. Heating the sample at 95 °C inactivates both Proteinase K and SARS-CoV-2 virions [43], thereby increasing the biosafety of the sample. It was critical to include a wash step to remove both RNAlater solution and saliva that were soaked up by the filter paper, as well as particulates bound to the filter paper. We designed LAMP primer sets throughout the SARS-CoV-2 genome (see Methods), and focused on the primer set (ZI-1, targeting ORF 1a) with the lowest predicted propensity for primer-dimer formation and that avoids known mutations among common SARS-CoV-2 variants that could affect primer binding specificity (Fig 2B and S1 Table for a list of SARS-CoV-2 mutations; covariants.org as of Aug 9, 2021]). To evaluate the specificity of the ZI-1 primers, we tested a panel of 22 inactivated respiratory pathogens, including SARS-CoV-1, MERS, H1N1 influenza, and common respiratory coronaviruses. We detected SARS-CoV-2 RNA in samples containing encapsulated SARS-CoV-2 RNA particles, but not in samples containing only the other respiratory pathogens (Fig 3A–3D), indicating that the primers were specific to SARS-CoV-2 (Fig 3A: 9/9 SARS-CoV-2 vs: 0/11 SARS-CoV-1 $P<0.0001$, vs 0/11 MERS $P<0.0001$; Fig 3D: 11/11 SARS-CoV-2 vs 0/11 replicas of each of 5 different respiratory pathogen pools, $P<0.0001$).

## Assay consistency

To determine the consistency of RNA extraction from saliva using WHotLAMP, we designed intron-spanning LAMP primers to detect human RAB7A mRNA, a transcript expressed at

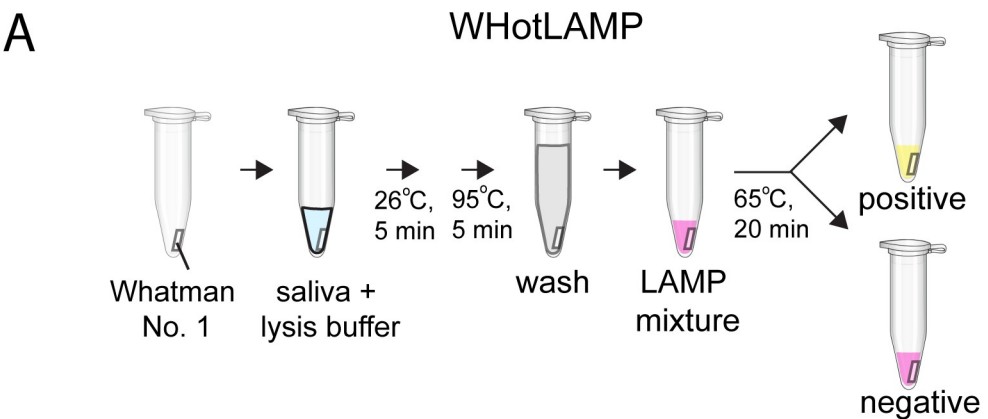

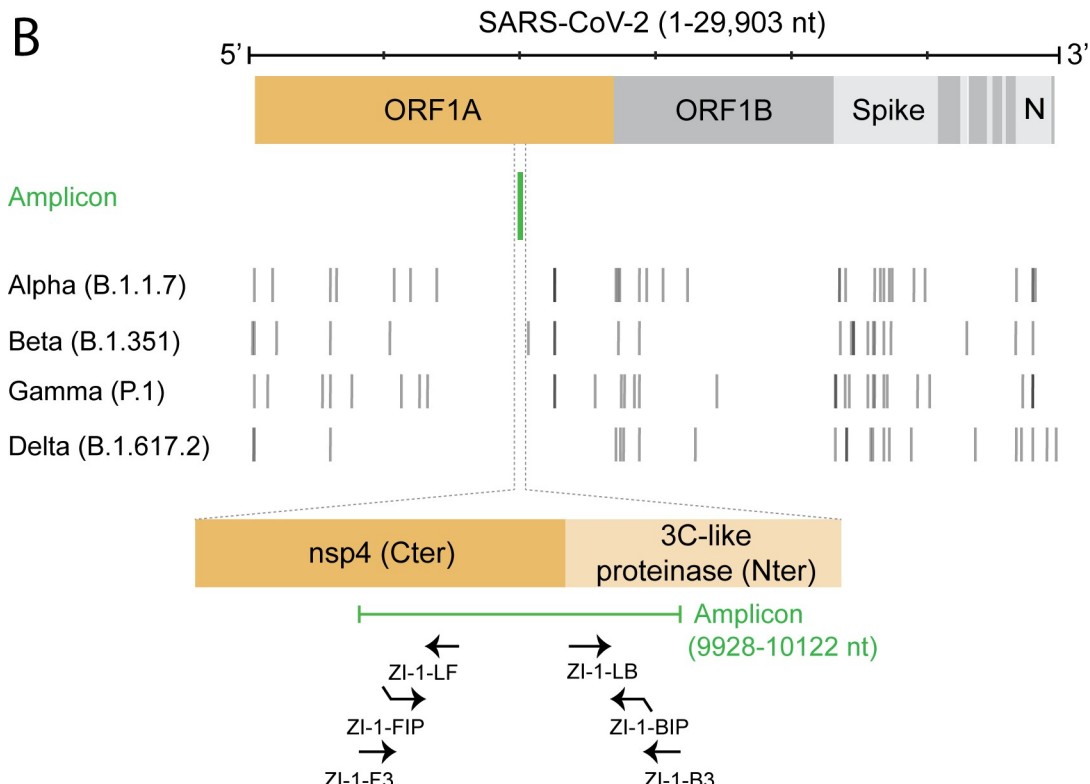

**Fig 2. Overview of WHotLAMP assay and primers. A**. Schematic of the WHotLAMP assay. **B**. Location of ZI-1 LAMP primers and amplicon relative to mutations (vertical lines) defining SARS-CoV-2 variants.

high levels in multiple tissues [44]. RAB7A LAMP primers led to a yellow color change with saliva from healthy donors, but not when RNase A was added after Proteinase K treatment, indicating the amplification originated from RNA and not genomic DNA (Fig 4A and 4B). In contrast, the reaction control (RPP30) in the CDC-recommended RT-qPCR test panel for SARS-CoV-2 amplifies both genomic DNA as well as cDNA [45]. Further testing of additional saliva samples with RAB7A LAMP led to a yellow color change in 20 of 20 individual samples, indicating that the RAB7A LAMP primers are an appropriate control for benchmarking successful RNA extractions from saliva.

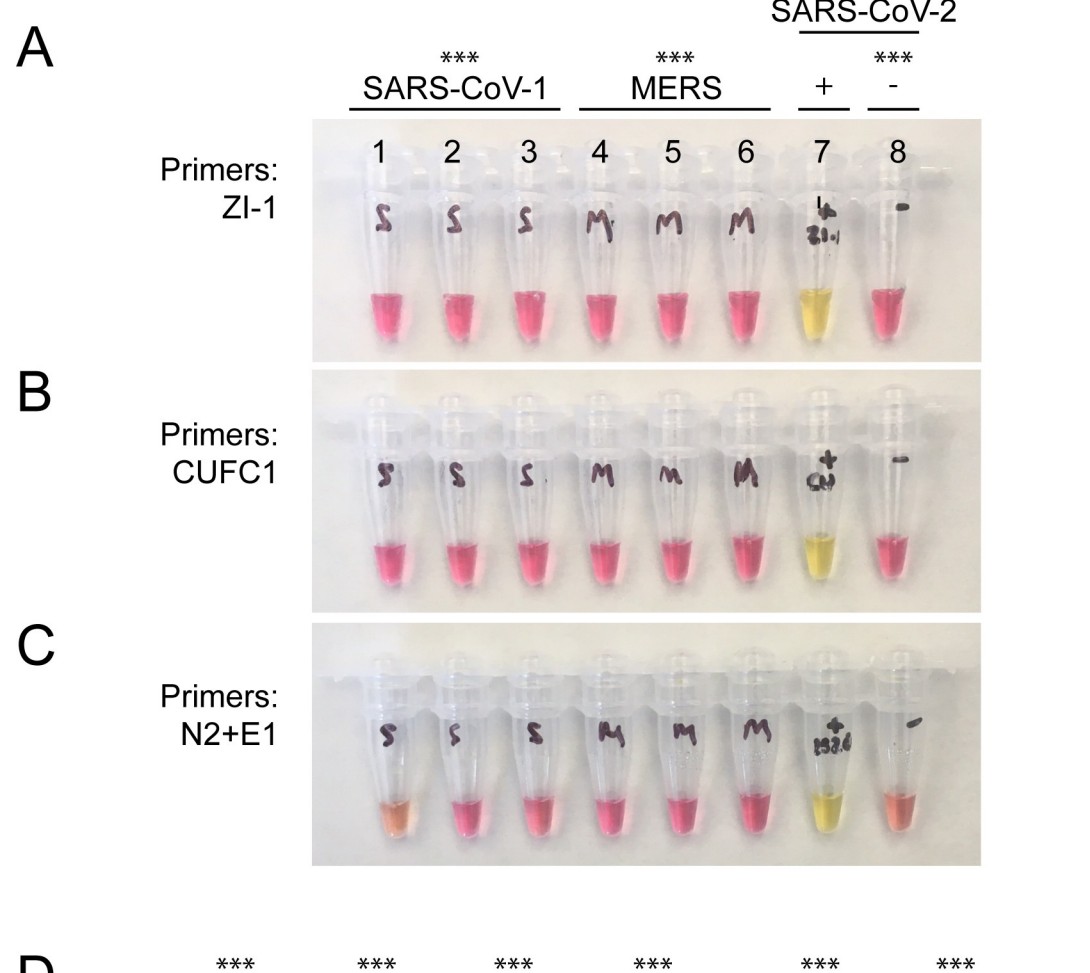

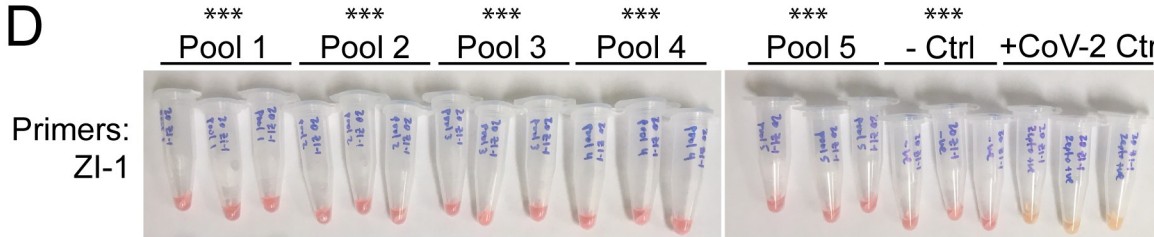

**Fig 3. Specificity of SARS-CoV-2 LAMP primers. A**. Representative LAMP reactions using ZI-1 LAMP primers with $1\times10^5$ copies of SARS-CoV-1 DNA (tubes 1–3), MERS DNA (tubes 4–6), SARS-CoV-2 RNA (tube 7), and no template control (tube 8). **B** and **C**, same as **A** but with CUFC1 or N2+E1 LAMP primers, respectively. **D**. Representative LAMP reactions with ZI-1 LAMP primers using WHotLAMP detecting different respiratory pathogens (Pools 1–5), no respiratory pathogens (- Ctrl), and with inactivated SARS-CoV-2 virions (+CoV-2 Ctrl). ***$P<0.0001$ vs. positive CoV-2 control by Fisher's exact test.

## Color variation among healthy donor saliva and automated scoring of assay

To devise a quantitative colorimetric threshold from which to differentiate between positive and negative LAMP results, we photographed LAMP assays under controlled illumination using a custom-made portable photobox (Fig 5A). A potential concern regarding testing saliva using pH-sensitive dyes, rests in how the variability of pH of saliva samples could influence the specificity of this test [46]. To examine the colorimetric variability of WHotLAMP, we tested

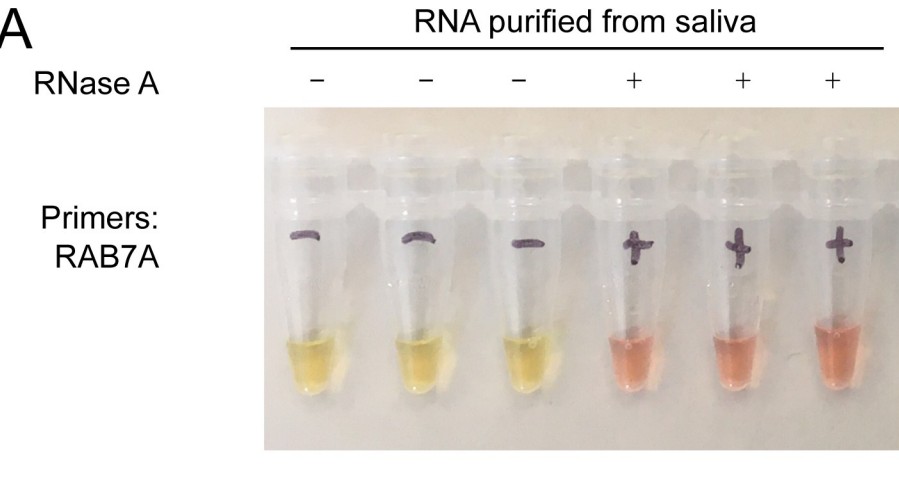

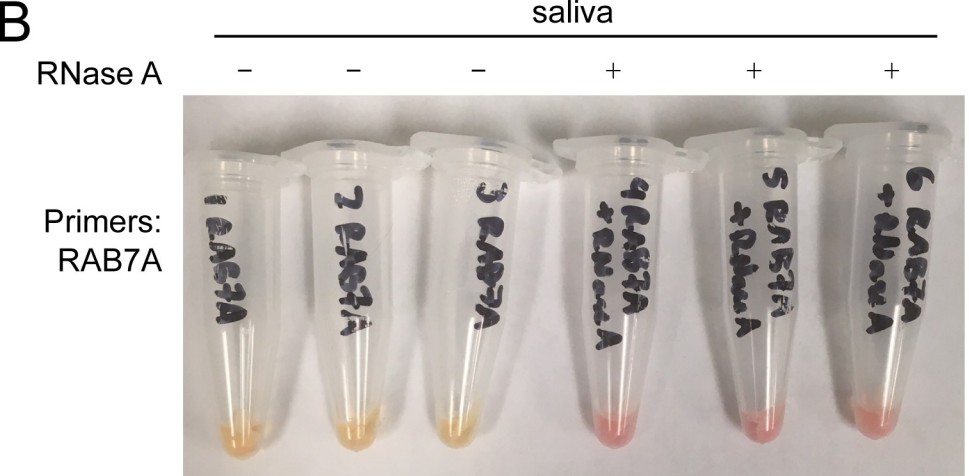

**Fig 4. Detection of RAB7A RNA in saliva. A**. LAMP reactions using RAB7A LAMP primers with purified RNA from healthy saliva (tubes 1–3), or purified RNA treated with RNase A (tubes 4–6). **B**. LAMP reactions using WHotLAMP detecting RAB7A in saliva (tubes 1–3), or with RNase A treatment (tubes 4–6).

saliva samples from 36 healthy volunteers (nasal swab SARS-CoV-2 qPCR negative) and saliva samples from people with a positive nasal swab SARS-CoV-2 qPCR. Using automatic image processing, we extracted the hue of each reaction (Fig 5B). The range of hues (Fig 5C) of healthy salivas did not overlap with the range of SARS-CoV-2 positive saliva samples (n = 36 negative samples vs. n = 67 positive samples, $P<0.0001$), indicating unambiguous colorimetric classification of results (Fig 5D).

## Limit of detection

We next performed a series of dilutions of a SARS-CoV-2 positive saliva sample to estimate the limit of detection (LoD). Using a saliva sample from an individual with a positive nasal swab Ct (threshold cycle) value of 21 using CDC nCoV N1 and N2 PCR primers to amplify the N gene, we could detect SARS-CoV-2 using WHotLAMP in 20/20 (100%) of replicate saliva samples diluted 1:20,000, and 19/20 (95%) of replicate samples diluted 1:40,000,

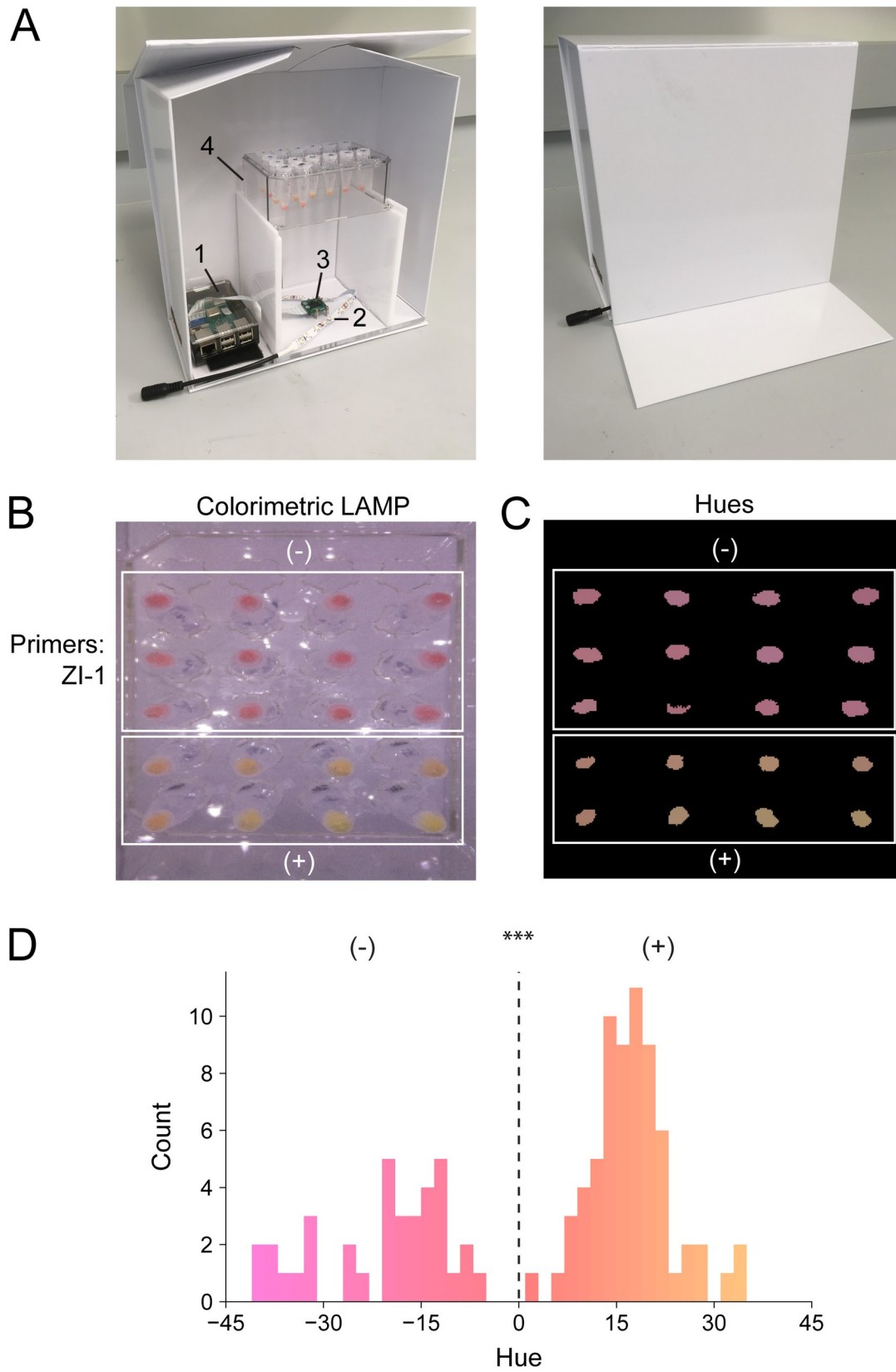

**Fig 5. Colorimetric quantification of LAMP reactions. A**. Illuminated lightbox with automated image acquisition using Raspberry Pi. 1) Raspberry Pi unit; 2) white LED strip; 3) camera unit; 4) test tube rack. **B**. LAMP reactions using WHotLAMP with ZI-1 primers on saliva samples from different negative nasal-swab qPCR SARS-CoV-2 individuals (top white box) and SARS-CoV-2 positive (nasal swab) samples (bottom white box). **C**. Processed image showing conversion of colorimetric LAMP results to hues. **D**. Hue distribution of WHotLAMP saliva results from negative (-) and positive (+) nasal-swab SARS-CoV-2 qPCR donor samples. ***P<0.0001 of negative vs. positive CoV-2 samples by *t*-test.

suggesting a LoD of Ct ~36. To better quantify the LoD of our assay, we purified RNA from the same saliva and performed RT-qPCR using CDC 2019 nCoV N1 and N2 PCR primers. Through interpolation to a standard curve ($R^2$ = 0.99) using a dilution series of a standard (IDT N-gene), we determined that the LoD of WHotLAMP with ZI-1 primers corresponds to ~3.6 viral N gene RNA copies/μL saliva. This LoD matches the 4 viral N gene RNA copies/μL of saliva determined using encapsulated SARS-CoV-2 RNA spiked into saliva (Fig 1B).

## Specificity and sensitivity

To evaluate the clinical sensitivity of WHotLAMP, we tested saliva from patients who at the same time tested positive for SARS-CoV-2 with a nasal swab qPCR (See S2 Table for patient information). WHotLAMP with ZI-1 primers detected 36/38 (94.7%) positives with a Ct value up to 34 (Fig 6A) and WHotLAMP positive tests had significantly lower Ct values (*P* = 0.0005), indicating a higher sensitivity of WHotLAMP for samples with higher viral load. In contrast, CUFC1 primers [31] detected only 25/32 (78.1%) positives amongst the same cohort of SARS-CoV-2 saliva samples (Fig 6B) and, as with ZI-1 primers, positive tests had significantly lower Ct values (*P* = 0.0025). Previous reports indicate low success in culturing SARS-CoV-2 from patients with a positive nasal swab at a Ct value >34 [22, 47], suggesting that the WHotLAMP assay with ZI-1 primers can detect nearly all individuals that carry viral loads considered to be contagious. Furthermore, while the most accurate antigen tests have a false negative rate of ~20% for samples with a Ct <30 [26], WHotLAMP with ZI-1 primers detected 21/21 samples that had Ct <31 in nasal swab RT-qPCR.

To evaluate the specificity of WHotLAMP, we tested saliva from asymptomatic individuals who had a negative qPCR result from a nasal swab taken within 24 hours of the saliva collection. Notably, we found no false positives among 40 individual samples (false-positive rate <1/40; CI = 0–0.091) using ZI-1 primers, indicating this primer set offers high specificity (Fig 6C), whereas CUFC1 primers detected 2 false positives out of 37 individual samples (a subset of the 40 tested with ZI-1; false-positive rate 0.054; CI = 0.0097–0.18) (Fig 6D).

## Concluding remarks

We describe WHotLAMP, a simple and inexpensive molecular test (~$3.00 for consumables per reaction at retail prices) that does not require specialized laboratory equipment, to detect SARS-CoV-2 virus in saliva. We show that WHotLAMP can detect low levels of SARS-CoV-2 virus in saliva in 30 minutes. Its low false-positive rate allows for deployment under conditions of low prevalence, where a high test specificity is particularly important to achieve high positive predictive values. The current assay design is already applicable to test at POC settings. Moreover, its single-tube format that requires no centrifugation, is conducive to scaling to 96-well formats, but can also be adapted for home use for frequent self-administered monitoring. While here we focused on a test for SARS-CoV-2, this technology could be used to detect other pathogens that are present in saliva by substituting primers [48], making WHotLAMP a broadly useful diagnostic assay.

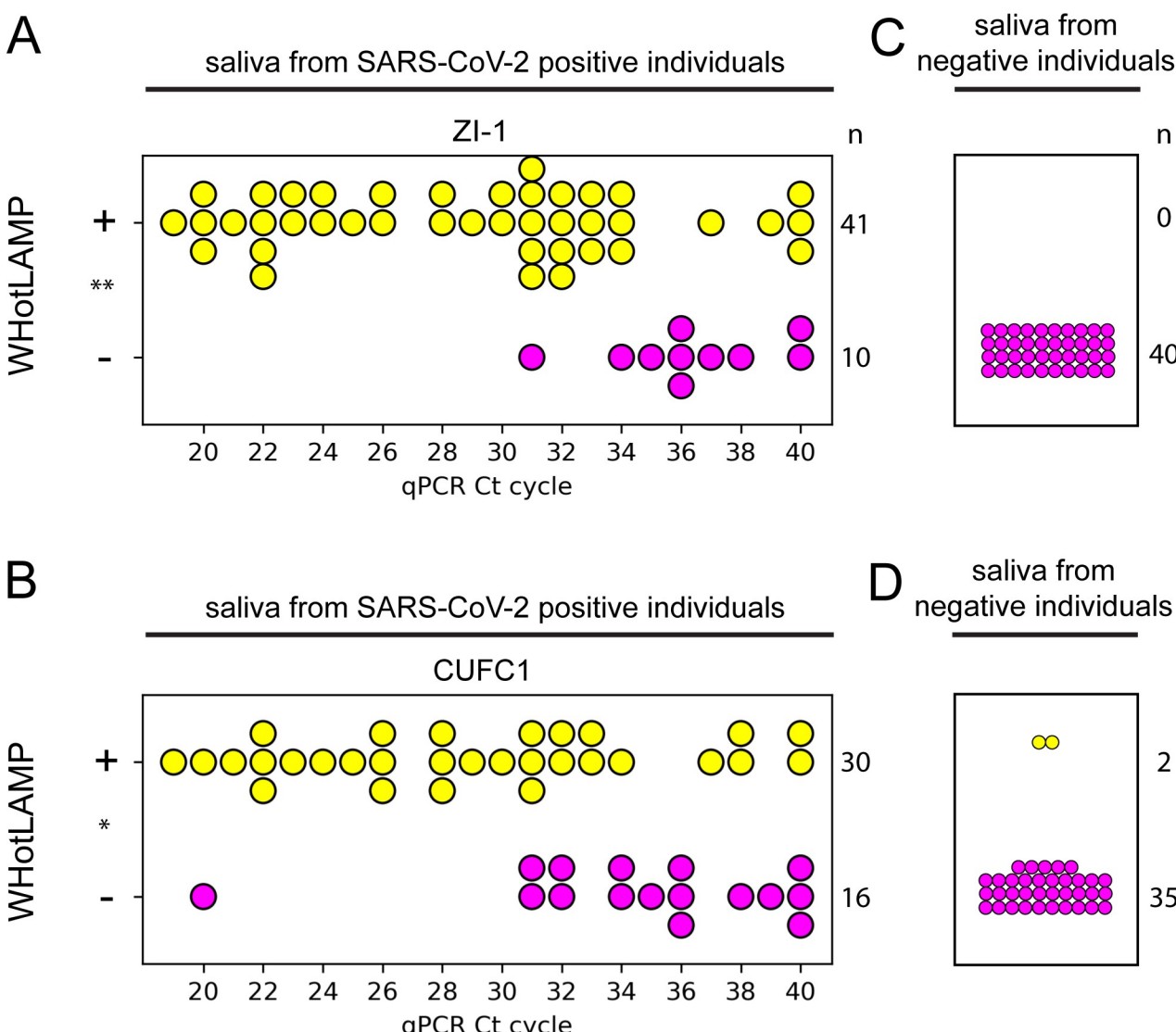

**Fig 6. Sensitivity and specificity of WHotLAMP. A**, **B** Sensitivity and **C**, **D** specificity of WHotLAMP using (**A**, **C**) ZI-1 or (**B**, **D**) CUFC1 primers with qPCR SARS-CoV-2 positive saliva. Yellow circles denote positive (+) LAMP reactions and magenta circles denote negative (-) LAMP reactions. ** *P*<0.001 and * *P*<0.01 between WHotLAMP positive and negative samples, by *t*-test.

## Methods

### Saliva RNA spike-in assay

For saliva RNA spike-in tests, a 1.7 mL tube with 100 μL of saliva was combined with 100 μL lysis buffer (0.8 M guanidine hydrochloride (G3272, Sigma), 2% Tween-20 (BP337, Fisher Biotech)), mixed, and incubated at room temperature for 5 min. Approximately $1x10^5$ copies of SARS-CoV-2 RNA (MT007544.1, TWIST Bioscience) were added to the saliva lysate and mixed. A 2x3 mm piece of Whatman No. 1 filter paper (Cat. 1001–929, GE Healthcare) was added to the lysate and incubated at room temperature for 1 min. The lysate was removed and the filter paper was washed twice. For each wash, 1 mL of wash buffer (1 mM Tris-Cl pH 8.0, 0.1 mM EDTA pH 8.0, 0.1% Tween-20) was added, inverted 20x, incubated for 1 minute at room temperature, and then removed. The filter paper was then transferred to a PCR tube containing 40 μL of 1x LAMP reaction mixture (see below).

For saliva encapsulated SARS-CoV-2 RNA spike-in tests, the extraction was performed as described for the RNA spike-in assay, except that encapsulated RNA controls (either AccuPlex SARS-CoV-2 or human RNaseP, 0505–0168, LGC Sera Care) were added to the saliva lysate mixture instead of naked RNA. For the no extraction control, encapsulated SARS-CoV-2 RNA control was added directly to saliva without treatment with lysis buffer.

## WHotLAMP assay

For WHotLAMP assays, a 1.7 mL tube with a piece of Whatman No.1 filter paper (approximately 6 mm$^2$ total surface area) affixed at the bottom using Kwik-Sil™ silicone (see below) was used to carry out the entire assay. For each sample, 100 μL of saliva was loaded into the tube, followed by 50 μL of RNAlater (R0901, Sigma) and 25 μL of Proteinase K (10 mg/mL) (PB0451, BioBasic; or 25530–049, Ambion), and mixed. The tube was incubated at 26 °C for 5 min, then at 95 °C for 5 min, and then returned to room temperature. The saliva mixture was removed and the filter paper was washed twice. For the first wash, 1 mL of wash buffer (1 mM Tris-Cl pH 8.0, 0.1 mM EDTA pH 8.0, 0.1% Tween-20) was added, inverted 20x, incubated for 1 min at room temperature, and then removed. For the second wash, 1 mL of wash buffer was added and the tube was incubated for 1 min at room temperature. The wash buffer was then removed, and 50 μL of 1x LAMP reaction mixture was added. All LAMP reactions consisted of 1x colorimetric RT-LAMP mixture (NEB M1800S), 40 mM guanidine hydrochloride (G3272, Sigma, freshly made) and LAMP primers (1.6 μM FIP/BIP, 0.2 μM F3/B3, 0.4 μM LF/LB). Reactions were carried out at 65 °C for 20 minutes (for 1.7 mL tubes) in a heat block or 45 minutes (for PCR tubes) in a thermocycler. Tubes were cooled afterwards to enhance color contrast between positive and negative colorimetric LAMP results.

## Assay to test for cross-reactivity of respiratory pathogens

ZI-1, CUFC1 and N2+E1 LAMP primers were used in 25 μL colorimetric RT-LAMP reactions conducted in PCR tubes. 1x10$^5$ DNA copies of SARS-CoV-1 and MERS, or RNA purified from positive SARS-CoV-2 saliva were added. Reactions were carried out at 65 °C for 45 min. To test whether ZI-1 primers had cross-reactivity with other respiratory pathogens, the WHotLAMP assay was followed, except that the 100 μL of saliva was substituted with 100 μL of respiratory pathogen mixture (20 μL of a respiratory control panel (NATRPP-1, ZeptoMetrix) or SARS-CoV-2 control (NATSARS(COV2)-ERC, ZeptoMetrix) with 80 μL water). ZeptoMetrix respiratory control panels contain the following non-infectious viral particles or bacterial cells: (Pool 1), Influenza A H1N1, Parainfluenza Type 4A and 4B, Rhinovirus (1A), Adenovirus Type 3; (Pool 2), Influenza A H1, Respiratory Syncytial Virus A, Parainfluenza Type 1, Coronavirus NL63, Mycoplasma pneumonia (M129); (Pool 3), Influenza A H3, Respiratory Syncytial Virus B, Coronavirus OC43, Coronavirus HKU-1; (Pool 4), Influenza B, Parainfluenza Type 3, Human Metapneumovirus, Legionella pneumophila; (Pool 5), Parainfluenza Type 2, Coronavirus 229E, Human Bocavirus, Chlamydophila pneumoniae.

## Ethics statement

Human subject research was approved by Columbia University IRB (protocol AAAT1974) and patients provided written informed consent.

## Collection and processing of patient nasal and saliva samples

Patients provided sterile cotton tipped swabs and conical tubes for sample collection. First, patients underwent separate nose and throat swabs for COVID-19 PCR analysis. For nose

swabs, patients were instructed to swab 10 circles per nostril at ~1–2 cm from the nasal opening. For throat samples, patients self-swabbed. Swabs were then placed into 500 μL RLT buffer (RNeasy Mini kit, 74106, Qiagen) with 10 μL beta-mercaptoethanol per 1 mL RLT buffer. Participants were instructed, if able, to swish and swallow a small amount of water to clean the mouth. Participants were then asked to produce saliva that naturally pools in their mouth (not expectorated) into a 50 mL Falcon sterile tube. All specimens were assayed within 2 hrs or stored at 4˚C for up to 48 hrs for further analysis.

For RNA isolation and real-time qPCR analysis, RNA was purified using the RNeasy Mini Kit (Qiagen) with minor modifications to the standard protocol: Both the nose and throat lysates were combined on a single column for RNA isolation. In addition, only 1 RPE buffer spin was performed, followed by an 80% ethanol spin. The membrane was dried at full speed centrifugation for 5 min. RNA was eluted with two separate 20 μL RNase-free water 5 min incubations and 1 min full speed spins. RNA was transcribed into cDNA utilizing the High-Capacity cDNA Reverse Transcription Kit (Applied Biosystems) with the following thermocycler settings: 25˚C for 10 min, 37˚C for 60 min, 85˚C for 5 min, then 4˚C until used.

Real-time qPCR was performed on cDNA according to standard protocols utilizing Taq-Path qPCR Master Mix, ThermoScientific Microamp 96-well reaction plates, and the QuantStudio 3 Real Time PCR system. COVID-19 N1 and N2 FAM primers (2019-nCoV_N1-P, FAM-ACC CCG CAT TAC GTT TGG TGG ACC-BHQ1; 2019-nCoV_N2-P, FAM-ACA ATT TGC CCC CAG CGC TTC AG-BHQ1) were analyzed out to 40 cycles as compared to 18S rRNA VIC loading control. A 20 μL reaction was performed with 10 μL master mix, 1 μL of COVID-19 N1 or N2 primer sequence, 1 μL of 18s rRNA endogenous control, 4 μL nuclease-free water, and 4 μL RNA were added to each well. All assays were run in duplicate. Each plate was run with a COVID-19 positive control (Integrated DNA Technologies, 2019-nCoV_N_Positive Control, #10006625), and water as a negative control. Samples were deemed negative if by qPCR there was no amplification for N1 or N2. The average Ct of the duplicates was used.

Saliva samples and WHotLAMP assays were handled and processed under BSL-2 containment. To test the sensitivity of WHotLAMP, patient saliva samples and negative control samples were tested under blind conditions. Saliva samples from healthy volunteers used to assess saliva variability were collected without prior food or beverage restrictions. Samples were tested using the WHotLAMP assay with 1.7 mL tubes.

## Purification of RNA from saliva for RAB7A LAMP and LoD qPCR

RNA was purified from saliva samples using RNeasy Mini columns (Qiagen). 250 μL of saliva was mixed with 250 μL of RLT buffer and 500 μL of 70% ethanol. 500 μL of the mixture was loaded onto a column and centrifuged at 14,000g for 30 s. A second 500 μL volume was loaded onto the same column and centrifuged. The column was washed with 500 μL of RPE, centrifuged, transferred to a new tube, and spun to dry. The column was transferred to a fresh tube and eluted in 20 μL of water. Two additional elutions using 20 μL of water were performed, and all eluates were pooled into one tube.

## cDNA synthesis for saliva LoD analysis

For cDNA synthesis, total RNA purified from saliva was reverse transcribed using random primers and recombinant M-MuLV reverse transcriptase (E6300S, NEB) according to manufacturer's instructions. Briefly, 50 ng of total RNA was mixed with random primers and denatured for 5 min at 70˚C, spun briefly, and placed on ice. M-MuLV reaction mix and M-MuLV enzyme were added to the mixture and incubated at 25˚C for 5 min, then incubated at 42˚C

for 1 hr and heat inactivated at 80˚C for 5 min. The cDNA was then stored at -20˚C until further use.

## qPCR for saliva LoD analysis

cDNA from SARS-CoV-2 positive saliva and positive control SARS-CoV-2 N gene DNA (2019-nCoV N positive control, IDT Cat. 10006625), were diluted in triplicate. qPCR was performed using CDC N1 and N2 gene primers, their respective fluorescent probes (2019-nCoV_N1 Probe: FAM-ACC CCG CAT TAC GTT TGG TGG ACC-BHQ1 and 2019-nCoV_N2-Probe: FAM-ACA ATT TGC CCC CAG CGC TTC AG-BHQ1) and Taqman Fast Advanced master mix (ThermoFisher, Cat. 4444551), and a QuantStudio 3 Real-Time PCR system (Applied Biosystems) using recommended CDC 2019 nCoV RT-PCR thermocycling parameters. Ct values falling within the linear amplification range were used to estimate SARS-CoV-2 copy number in the cDNA, and an estimated LoD based on copy number was calculated based on the dilutions.

## RT-LAMP primers

Primers (desalted, Integrated DNA Technologies and Eurofins Genomics) used for LAMP reactions were prepared in water as 10x stocks (16 μM FIP/BIP, 2 μM F3/B3, 4 μM LF/LB) (see Table 2). ZI LAMP primer sets were identified using the NEB Primer Design Tool (https://lamp.neb.com/#!/) and a ~800 bp sliding window across the SARS-CoV-2 genomic sequence (MN908947.3). Primer sets with low primer ΔG values (e.g. <-2.2) were selected for further analysis. One primer set, ZI-1, was selected because it was highly sensitive when tested using positive SARS-CoV-2 saliva.

## WHotLAMP LoD assay

SARS-CoV-2 positive saliva was initially heat-inactivated at 65 °C for 30 min prior to diluting using negative control saliva. An initial series using 10-fold dilutions was tested using WHotLAMP with ZI-1 primers. A second series using 2-fold dilutions was tested and repeated 20 times to estimate the consistency at the LoD. Saliva from the same SARS-CoV-2 positive sample was used to extract RNA for RT-qPCR analysis.

## Adhesive and material handling and testing

Adhesives that could be used to affix Whatman No. 1 filter paper to the bottom of a tube were tested to determine if they were compatible with the LAMP colorimetric assay. A small quantity of liquid adhesives (~2 μL) was spotted at the bottom of a tube and allowed to cure for at least 24 hrs. For solid materials, ~1–2 mm² pieces were used. Tubes containing different adhesives were incubating in 1x LAMP reaction mixture at 65 °C for 20 min (for 1.7 mL tubes) or 45 min (for 0.2 mL PCR tubes). Materials tested were, silicones (Kwik-Sil™ silicone elastomer [World Precision Instruments], aquarium silicone [Aqueon], neutral cure silicone [Dow Corning 737]), liquid glues (Gorilla glue [original, polyurethane], Crazy Glue [Loctite]), glue gun sticks (polyamide [Power Adhesives TEC Bond 7718], acrylic [Infinity Bond], Hot Melt Mini Glue Sticks [Allary]), rubber cement (paper cement [Best-Test]), double-sided tape (Permanent [Scotch, 3M]), plastic strip, (cut from a 175 micron polyester sheet [Grafix Plastics]). To prepare large numbers of 1.7 mL tubes with Whatman No. 1 filter paper glued with Kwik-Sil™, a small aliquot of component A and component B were mixed together and placed on ice to slow the polymerization process. Tear-shaped Whatman filter paper pieces were prepared using a hole-punch. The tapered end of the paper was dipped slightly in Kwik-Sil™ and placed,

**Table 2. Primers used in study.**

| Primer | Sequence |
|---|---|
| ZI-1-F3 | 5' GGA TAC AAC TAG CTA CAG AGA A 3' |
| ZI-1-B3 | CCA CAA GTT ACT TGT ACC ATA C |
| ZI-1-FIP | TTG GTA AAG AAC ATC AGA ACC TGA GGC TGC TTG TTG TCA TCT C |
| ZI-1-BIP | CCA CCA CAA ACC TCT ATC ACC TAA CCC TCA ACT TTA CCA GAT |
| ZI-1-LF | AAG TCA TTG AGA GCC TTT GC |
| ZI-1-LB | GTG GTT TTA GAA AAA TGG CAT TCC C |
| CUFC1-F3 | TGG ATA CAA CTA GCT ACA GAG AAG |
| CUFC1-B3 | AGC CAA AGA CCG TTA AGT GTA |
| CUFC1-FIP | GTG GTG GTT GGT AAA GAA CAT CAG ACT TGT TGT CAT CTC GCA AAG G |
| CUFC1-BIP | CCT CTA TCA CCT CAG CTG TTT TGC TGT ACC ATA CAA CCC TCA ACT T |
| CUFC1-LF | ACC TGA GTT ACT GAA GTC ATT GAG A |
| CUFC1-LB | TGG TTT TAG AAA AAT GGC ATT CCC |
| N2-F3 | ACC AGG AAC TAA TCA GAC AAG |
| N2-B3 | GAC TTG ATC TTT GAA ATT TGG ATCT |
| N2-FIP | TTC CGA AGA ACG CTG AAG CGG AAC TGA TTA CAA ACA TTG GCC |
| N2-BIP | CGC ATT GGC ATG GAA GTC ACA ATT TGA TGG CAC CTG TGT A |
| N2-LF | GGG GGC AAA TTG TGC AAT TTG |
| N2-LB | CTT CGG GAA CGT GGT TGA CC |
| E1-F3 | TGA GTA CGA ACT TAT GTA CTC AT |
| E1-B3 | TTC AGA TTT TTA ACA CGA GAG T |
| E1-FIP | ACC ACG AAA GCA AGA AAA AGA AGT TCG TTT CGG AAG AGA CAG |
| E1-BIP | TTG CTA GTT ACA CTA GCC ATC CTT AGG TTT TAC AAG ACT CAC GT |
| E1-LB | GCG CTT CGA TTG TGT GCG T |
| E1-LF | CGC TAT TAA CTA TTA ACG |
| RAB7A-F3 | ACA GGC CTG GTG CTA CAG |
| RAB7A-B3 | CTG CAG CTT TCT GCC GAG |
| RAB7A-FIP | CAA TCG TCT GGA ACG CCT GCT CCC TAC TTT GAG ACC AGT GC |
| RAB7A-BIP | AAG CAG GAA ACG GAG GTG GAG GCC CGG TCA TTC TTG TCC |
| RAB7A-LF | ACG TTG ATG GCC TCC TTG |
| RAB7A-LB | TGT ACA ACG AAT TTC CTG AAC C |
| 2019-nCoV_N1-F | GAC CCC AAA ATC AGC GAA AT |
| 2019-nCoV_N1-R | TCT GGT TAC TGC CAG TTG AAT CTG |
| 2019-nCoV_N1-P | FAM-ACC CCG CAT TAC GTT TGG TGG ACC-BHQ1 |
| 2019-nCoV_N2-F | TTA CAA ACA TTG GCC GCA AA |
| 2019-nCoV_N2-R | GCG CGA CAT TCC GAA GAA |
| 2019-nCoV_N2-P | FAM-ACA ATT TGC CCC CAG CGC TTC AG-BHQ1 |

taper side facing up, at the bottom of the 1.7 mL tube using fine-tipped forceps, and air dried for at least 24 hrs.

## Raspberry Pi lightbox

To quantify colorimetric ranges under uniform conditions we embedded an enclosed white box with: 1) a Raspberry Pi 3 (model B+), 2) camera unit (camera v2.1), and 3) white LED lights (DC12V LED strip). The raspistill command line tool was run to capture still images (raspistill—raw -o png). A color chart (Digital Kolor Kard) inside the box was used as reference to calibrate the white balance of images. Images were then used to extract hues to interpret WHotLAMP positive and negative colorimetric results.

## Image processing

We developed a proof-of-concept image analysis pipeline that identified sample results. Images of an array of samples were acquired and thresholded based on color saturation to identify regions of interest (ROIs) corresponding to samples. Areas of high or low brightness as well as areas near the image border were excluded from potential ROIs. We found that this method successfully identified correct ROIs and that the average hue within each ROI formed a bimodal distribution that could be used to successfully categorize samples into positives and negatives.

## Statistical analysis

Statistical analyses of colorimetric data were performed using Fisher's exact tests for frequencies (Fig 3A and 3D) and a *t*-test for continues hue values (Figs 5D, 6A and 6B). Sample sizes were: Fig 3A: n = 9 positives, n = 11 for each negative control group; Fig 3D, n = 9 positives and n = 9 for each negative control pool; Fig 5D, n = 36 negative samples, n = 67 nasal qPCR positive SARS-CoV-2 saliva samples; Fig 6A, n = 51 nasal qPCR positive SARS-CoV-2 saliva samples; Fig 5B, n = 46 nasal qPCR positive SARS-CoV-2 saliva samples.

## Supporting information

**S1 Table. SARS-CoV-2 mutations.**
(XLSX)

**S2 Table. Patient information.**
(XLSX)

**S1 File. WHotLAMP protocol.**
(PDF)

**S2 File. Python code for LAMP colorimetric result analysis.**
(PDF)

## Acknowledgments

Natalie Steineman, Marjorie Xie, and Aniruddah Das provided organizational support. Julian Valdes provided input on LAMP primer design. The Zuckerman Institute Scientific Platforms provided resources. We thank Rui Costa and the Zuckerman Institute for unwavering support.

## Author Contributions

**Conceptualization:** David Ng, Ana Pinharanda, Merly C. Vogt, Darcy S. Peterka, Tanya Tabachnik, Andres Bendesky.

**Data curation:** Monica Goldklang, Andres Bendesky.

**Formal analysis:** David Ng, Ashok Litwin-Kumar, Monica Goldklang, Andres Bendesky.

**Funding acquisition:** Peter Andolfatto, Andres Bendesky.

**Investigation:** David Ng, Ana Pinharanda, Merly C. Vogt, Kyle Stearns, Urvashi Thopte, Enrico Cannavo, Felix Fiederling, Stylianos Kosmidis, Barbara Noro, Ines Rodrigues-Vaz, Hani Shayya, Jeanine D'Armiento, Monica Goldklang.

**Methodology:** David Ng, Ana Pinharanda, Merly C. Vogt, Enrico Cannavo, Felix Fiederling, Stylianos Kosmidis, Barbara Noro, Hani Shayya, Andres Bendesky.

**Resources:** Kyle Stearns, Armen Enikolopov, Tanya Tabachnik, Jeanine D'Armiento, Monica Goldklang.

**Software:** Ashok Litwin-Kumar, Andres Bendesky.

**Supervision:** David Ng, Jeanine D'Armiento, Andres Bendesky.

**Validation:** David Ng, Ana Pinharanda, Merly C. Vogt, Urvashi Thopte.

**Visualization:** David Ng, Andres Bendesky.

**Writing – original draft:** David Ng, Ashok Litwin-Kumar, Monica Goldklang, Andres Bendesky.

**Writing – review & editing:** David Ng, Ana Pinharanda, Merly C. Vogt, Ashok Litwin-Kumar, Jeanine D'Armiento, Monica Goldklang, Andres Bendesky.

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
