## [Decision Letter · Decision Letter 0]

4 Aug 2021

PONE-D-21-23691

WHotLAMP: A simple, inexpensive, and sensitive molecular test for the detection of SARS-CoV-2 in saliva

PLOS ONE

Dear Dr. Bendesky,

Thank you for submitting your manuscript to PLOS ONE. After careful consideration, we feel that it has merit but does not fully meet PLOS ONE’s publication criteria as it currently stands. Therefore, we invite you to submit a revised version of the manuscript that addresses the points raised during the review process.

Please revise the point about statistics.

We look forward to receiving your revised manuscript.

Kind regards,

Etsuro Ito

Academic Editor

PLOS ONE

Journal Requirements:

2. We note that you have a patent relating to material pertinent to this article. Please provide an amended statement of Competing Interests to declare this patent (Patent application 63/088,694), along with any other relevant declarations relating to employment, consultancy, patents, products in development or modified products etc. Please confirm that this does not alter your adherence to all PLOS ONE policies on sharing data and materials, as detailed online in our guide for authors http://journals.plos.org/plosone/s/competing-interests by including the following statement: "This does not alter our adherence to  PLOS ONE policies on sharing data and materials.” If there are restrictions on sharing of data and/or materials, please state these. Please note that we cannot proceed with consideration of your article until this information has been declared.

Reviewers' comments:

Reviewer's Responses to Questions

**Comments to the Author**

1. Is the manuscript technically sound, and do the data support the conclusions?

Reviewer #1: Yes

Reviewer #2: Yes

2. Has the statistical analysis been performed appropriately and rigorously? 

Reviewer #1: Yes

Reviewer #2: I Don't Know

3. Have the authors made all data underlying the findings in their manuscript fully available?

Reviewer #1: Yes

Reviewer #2: Yes

4. Is the manuscript presented in an intelligible fashion and written in standard English?

Reviewer #1: Yes

Reviewer #2: Yes

5. Review Comments to the Author

Reviewer #1: The WHotLAMP manuscript from Ng et al presents a very good colorimetric LAMP assay for SARS-CoV-2 detection. There have been quite a few of these published now but the performance, simplicity, and novelty in this approach justify publication. The manuscript is clear and well written, and I feel it can be accepted mostly as is after addressing a few comments listed below.

1) The authors use an Orf1 primer set, it would be good to note whether the targeted region is anywhere close to the SGF deletion found in alpha, beta, gamma, and lambda variants which may affect performance. Related, the LOD is measured using quantitative PCR with N gene primers, a comment on the veracity of the assumption that 1 N=1 Orf1 would be good.

2) The authors describe spiking naked control RNA into saliva and then using it for detection. Other studies and conventional wisdom would say this wouldn't work and that the RNA is degraded extremely rapidly. Is there a reason it works here w/o RNase inactivation?

3) In describing the LoD the authors say that their assay is ">50X more sensitive than other recent saliva LAMP assays" which is a bit unfair considering this is contrived samples, data in Fig. 5A doesn't really indicate a "50X" sensitivity improvement. Plus there are a lot of LAMP assays, picking 1 for LOD comparison doesn't quite justify a claim of superiority to "other assays".

Reviewer #2: Ng et al. have performed an interesting and clear study that may benefit the fight against the Covid-19 pandemic. The paper is clearly written and details a novel test for detecting the covid-19 spike protein in what appears to be a specific assay. The benefit of the assay is how easy it is to handle and that it can be translated to something that can be done at home, which allows more people to access a safe and specific covid-19 assay.

Major comments:

After reading the manuscript through and through several times, I can only make out one statistical test, however what test this was used is not mentioned. The Paper would greatly benefit from adding statistics throughout the paper and the statistics should be described in detail in the material and methods.

It is not clear how many samples that has been analyzed, there is a mention of 20/20 positive samples and then 21/21. A table of the patient/sample characteristics detailing the number of positive samples, the severity of disease, negative controls and other diseases would greatly benefit the paper.

6. PLOS authors have the option to publish the peer review history of their article (what does this mean?). If published, this will include your full peer review and any attached files.

Reviewer #1: No

Reviewer #2: **Yes: **Daniel Butler

---

## [Author Response · Author response to Decision Letter 0]

31 Aug 2021

We appreciate the thoughtful reviews and the opportunity to respond to the Reviewers’ concerns and suggestions. As detailed below, we have addressed all points by adding new data and statistical tests, new analyses and figures, and rewriting sections of the manuscript to improve clarity.

Reviewer 1:

1) The authors use an Orf1 primer set, it would be good to note whether the targeted region is anywhere close to the SGF deletion found in alpha, beta, gamma, and lambda variants which may affect performance. 

We have added a new figure panel (Figure 2B) that shows how our LAMP amplicon does not overlap any mutations in the alpha, beta, gamma, nor delta variants, suggesting our assay should be able to detect all these SARS-CoV-2 variants.

Related, the LOD is measured using quantitative PCR with N gene primers, a comment on the veracity of the assumption that 1 N=1 Orf1 would be good.

While SARS-CoV-2 virions package a single RNA genome, it is true that different SARS-CoV-2 genes are present at different levels in infected cells (Kim et al., 2020). In Figure 1B, we quantified the sensitivity of our assay using contrived saliva samples spiked with synthetic virions each containing single RNA viral copies (so each viral gene is 1 copy) and using LAMP primers targeting the N and E genes. This showed that we could detect as few as 4 viral particles per microliter of saliva (Fig 1A). As a more realistic measure of the limit of detection of our assay, we quantified the number of N gene RNA copies in saliva from COVID-19 patients through qPCR using the CDC approved N gene primers. This, followed by serial dilutions of these saliva samples showed that our LAMP assay has a limit of detection of ~3.6 N gene RNA copies per microliter of saliva. We have rewritten the text so it is clear that our limit of detection in saliva with people with COVID-19 is measured in terms of RNA copies of the N gene.

2) The authors describe spiking naked control RNA into saliva and then using it for detection. Other studies and conventional wisdom would say this wouldn't work and that the RNA is degraded extremely rapidly. Is there a reason it works here w/o RNase inactivation?

We agree with the reviewer that because of the high level of RNases present in saliva, direct spike-in of naked RNA into saliva would lead to rapid degradation, as we (data not shown) and others have observed. In this assay (Figure 1A in the original manuscript), our goal was to determine whether Whatman paper had the potential to capture SARS-CoV-2 RNA in saliva. We would like to point out to that in this assay, saliva was initially mixed and pre-incubated with a lysis buffer with a final concentration of 0.4 M guanidine hydrochloride (GH) before the addition of naked RNA. Although the concentration of GH is insufficient to denature and fully inactivate RNases, it has been reported that low concentrations of GH can partially inactivate RNaseA very rapidly. For example at a concentration of 0.5 M GH, about 50% of RNaseA activity is lost (Liu and Tsou, 1987). While the precise species of RNase(s) in the saliva samples is not known, this study supports the notion that pre-treatment with GH inhibits some ribonuclease activity.

We suggest that under the conditions used in this assay, there is partial inactivation of RNases in the saliva sample during the pre-incubation step. Combined with the high number (100,000) of RNA copies added and brief incubation time with Whatman paper (~1 min) before washing, sufficient RNA target is bound to the filter paper to be detected using LAMP. Because RNases are likely only partially inactivated under these conditions, we suspect that longer exposure times of naked RNA to saliva will lead to the complete degradation of the RNA target. The details of the conditions used in this assay are present in the Methods section in the original manuscript (lines 203-211), including composition of the lysis buffer and incubation times.

Of note, we performed these experiments using naked RNA spiked into saliva only in Fig 1A as a proof of principle that RNA can bind the Whatman paper in the context of saliva. In Fig 1B we use synthetic (encapsulated RNA) viral particles; in subsequent Figures using saliva, samples were incubated with RNAlater to inhibit RNase activity. In Fig 4 we show how we can detect endogenous human mRNA (as positive control of the assay); and in Fig 5 and Fig 6 we show how our assay detects SARS-CoV-2 RNA in real COVID-19 patients.

Kim, D., Lee, J.-Y., Yang, J.-S., Kim, J.W., Kim, V.N, and Chang, H. The Architecture of SARS-CoV-2 Transcriptome. Cell 181:914-921 (2020).

Liu, W., and Tsou, C.L. Activity change during unfolding of bovine pancreatic ribonuclease A in guanidine. Biochimica et Biophysica Acta 916:455-464 (1987).

3) In describing the LoD the authors say that their assay is ">50X more sensitive than other recent saliva LAMP assays" which is a bit unfair considering this is contrived samples, data in Fig. 5A doesn't really indicate a "50X" sensitivity improvement. Plus there are a lot of LAMP assays, picking 1 for LOD comparison doesn't quite justify a claim of superiority to "other assays".

We have revised this section according to this comment to not refer to a 50X improvement in sensitivity.

Reviewer 2:

The Paper would greatly benefit from adding statistics throughout the paper and the statistics should be described in detail in the material and methods.

Thank you for the suggestion. We have now increased our sample sizes in many parts of the paper and added statistical tests throughout (in main text and in figures), all of which support our observations and previous conclusions. We have a new section in the Methods explaining the statistical tests and the sample sizes. We also revised the data in the automatic quantification of WHotLAMP results using image analysis to remove some technical replicates. This removal did not change the non-overlapping distributions of hues between positive and negative samples and this difference in distributions in highly significant (P<0.0001).

It is not clear how many samples that has been analyzed, there is a mention of 20/20 positive samples and then 21/21. A table of the patient/sample characteristics detailing the number of positive samples, the severity of disease, negative controls and other diseases would greatly benefit the paper.

We have included a new Supplementary Table with details on numbers of COVID-19 patients tested, the severity of disease and other clinical details. We have also clarified throughout the revised manuscript the number of samples used in each experiment and also in a new section on Statistical analysis in the Methods.

---

## [Editor Report · Decision Letter 1]

2 Sep 2021

WHotLAMP: A simple, inexpensive, and sensitive molecular test for the detection of SARS-CoV-2 in saliva

PONE-D-21-23691R1

Dear Dr. Bendesky,

We’re pleased to inform you that your manuscript has been judged scientifically suitable for publication and will be formally accepted for publication once it meets all outstanding technical requirements.

Kind regards,

Etsuro Ito

Academic Editor

PLOS ONE

---

## [Editor Report · Acceptance letter]

7 Sep 2021

PONE-D-21-23691R1 

WHotLAMP: A simple, inexpensive, and sensitive molecular test for the detection of SARS-CoV-2 in saliva 

Dear Dr. Bendesky:

I'm pleased to inform you that your manuscript has been deemed suitable for publication in PLOS ONE. Congratulations! Your manuscript is now with our production department. 

Kind regards, 

on behalf of

Prof. Etsuro Ito 

Academic Editor

PLOS ONE